# Dynamics of Coastal Aquifers: Conceptualization and Steady-State Calibration of Multilayer Aquifer System—Southern Coast of Emilia Romagna

Claudia Cherubini [1,*] , Sadhasivam Sathish [1] and Nicola Pastore [2]

1   Department of Environmental and Prevention Sciences, University of Ferrara, 44122 Ferrara, Italy
2   Department of Civil, Environmental, Land, Construction and Chemistry Engineering, Polytechnical University of Bari, 70126 Bari, Italy; nicola.pastore@poliba.it
*   Correspondence: claudia.cherubini@unife.it

**Abstract:** Worldwide, coastal aquifers have been heavily exploited by socio economic activities for several decades, and climate change and sea level rise have also been threatening coastal aquifers. The authorities and policymakers have been advised to find the solutions in order to achieve sustainable water resources management. The southern part of Po delta, Italy is a low-lying coastal area also experiencing tectonic activity. Along with low-lying topography, unstable shore line and sea level, the groundwater is heavily exploited by this deltaic multilayered system of aquifers. Hence, a multilayer three-dimensional model of this aquifer system has allowed for the investigation of the response of aquifer to natural and anthropogenic exploitation. The present work regards the conceptualization of the multilayer aquifer system using lithological cross-sections, surface water features, and appropriate boundary conditions and the steady-state flow modelling. The spatially distributed elevations of the groundwater table and piezometric head from the different aquifers have been calibrated. The values of model error statistics at a satisfactory range, such as R-squared, mean error, root-mean-squared error and model efficiency, confirm that the developed model is reliable, and calibration is obtained with good match between observed and simulated data. The developed model can be used as a decision-making tool for the authorities and policymakers in order to plan for sustainable water management.

**Keywords:** groundwater modelling; coastal aquifer; Po delta; seawater intrusion; steady-state calibration

## 1. Introduction

Nowadays, climate changes have added extra stresses to subsurface water reservoirs, especially in the Mediterranean areas, due to an increase in freshwater demand for a variety of water uses and activities and a decrease in recharge. Drought and over-abstractions can significantly impact groundwater level recovery and groundwater quality long after droughts occur [1].

A publication by Eva Boergens in *Geophysical Research Letters* from the year 2020 [2] reported that there was a striking water shortage in Central Europe during the summer months of 2018 and 2019. The effects of this prolonged drought were evident in Europe in the summer of 2022.

Northern Italy is actually facing the worst drought of the last 70 years. Vast areas of the Po—the country's longest river that nourishes several northern and central regions—are already parched, while the winter water level of Lake Garda was the lowest in 35 years. As several aquifers and wells dry up, large areas are experiencing extreme water shortages.

Particularly coastal aquifers, irrespective of any landforms and geographic locations, are highly vulnerable to groundwater extraction and climate change impacts, such as sea level rise [3]. The recent global water cycle observations in the sixth assessment report (AR6) of climate change demonstrated extended droughts and increment in the frequency

and intensity of rainfall over the Mediterranean areas [4]. Disregarding the uncertainties in data acquisition, the known observations (at high confidence) and their predictions of each scenario still prove the water crisis in each sector because of their growing demand and shrinking of resources.

In this context, linking groundwater modeling with drought policy is needed to improve water management. Given the long-lasting impacts that drought may have on groundwater, predicting future droughts and identifying future uses and management priorities are all necessary [1].

Many researches have employed numerical modelling of groundwater flow in coastal environments.

Already in 1994, Nativ and Weisbrod [5], for the purpose of improving groundwater management, carried out a flow model of the Coastal Plain aquifer in Israel, which is heavily contaminated by agrochemicals and domestic and industrial waste. The aquifer had been traditionally managed as a single water reservoir. Their results indicated that the prevailing conceptualization of the aquifer as one homogeneous water body was incorrect; thus, the management of the Coastal Plain aquifer as a single reservoir was also improper; water in the upper sand unit appeared to circulate faster than in the deeper units, but its quality might have been inferior, probably due to anthropogenic effects.

Pouliaris et al. [6] developed a groundwater flow model for a coastal multilayer semi-arid aquifer system (Lavrio, Greece) with an uppermost alluvial granular aquifer and the lowermost karstic aquifer for the purpose of groundwater management. The sensitivity analysis and parameter estimation of the model parameters were conducted using a statistical approach, and the results showed that the head-dependent boundary condition could produce a more representative simulation of the coastal system hydrodynamics. They concluded that karstic aquifers can be simulated with conventional MODFLOW 2005 approaches; yet, an explicit insight to the karstic processes is not possible with this code and more sophisticated methodologies are necessary (MODFLOW CFP).

Priyanka and Kumar [7] developed a three-dimensional model of a coastal phreatic aquifer on the west coast of India by considering varying aquifer thickness and anisotropic heterogeneous aquifer parameters. The upscaled 3D model output for both state variables (h and C) were compared with a transversely isotropic model output that was developed using pumping test data. The mean temporal and spatial bias error and root-mean-squared error of the transversely isotropic model were greater than the upscaled model. Therefore, they concluded that upscaled conceptual 3D model was better than the transversely isotropic model.

Ranjbar et al. [8–10] developed an integrated framework for the management of coastal aquifers by developing a meta-model-based coupled simulation–optimization approach based on different machine learning algorithms as surrogate models for SEAWAT to accurately simulate the groundwater response to different pumping and recharge scenarios in two different aquifers in Iran.

Cherubini and Pastore [11] combined a density-driven, flow numerical model with a fault conceptual and hydrologic model to simulate different pumping patterns for the deep and the shallow aquifers of the Salento area (southeastern Italy), and thus defined critical stress scenarios for both aquifers. They proposed a solution strategy to protect the aquifers based on reduction in well density (number of pumping wells per unit area) coupled with artificial recharge.

Having 95% of the circumference surrounded by sea, Italy has a wide range of coastal landforms and low-lying polder areas. The Po delta, (northeastern Italy) is one of the low-lying areas characterized by a polder environment at the subaqueous delta. The water-logged and submerged area at the sparse subaqueous delta is reclaimed, and further water logging is controlled all the time by lowering the groundwater table using artificial drainage networks. The majority of the inland area is a flat surface with an average elevation of two meters below the Adriatic Sea level. The entire region experiences the impacts of climate change, sea level rise, groundwater pumping from multilayer aquifers,

storm surge, long periods of drought, soil erosion, shoreline alteration, land subsidence (caused by groundwater pumping and natural gas exploration), land use, artificial drains, morphometric alterations and groundwater tables below mean sea level [12,13].

The regional investigation of an area covering the central and northern part of the delta conducted by Teatini et al. [14] estimated spatially variable land subsidence caused by groundwater pumping, gas exploration and soil erosion. Antonioli et al. [15] provided a detailed account of regional sea level rise using tidal fluctuation data. Perini et al. [16] investigated the sea level rise along the coast of the Po delta and the resulting zone of flooding and land subsidence. The Water Framework Directive (2000/60/EC) indicates that the water basin authorities must utilize groundwater model tools for resource management practices [17] for the purpose of systematically updating the model and using it as a powerful tool for the sustainable management of water resources. Hence, Agenzia Prevenzione Ambiente Energia Emilia-Romagna and the Emilia-Romagna Region in the context of its water resources management started the development and the implementation of mathematical modeling for the groundwater system from the year 2001. The groundwater model was developed to include aspects such as regional water balance, feasibility of artificial recharge, nitrate transport and soil compaction for an area between 1.6 km$^2$ to 426 km$^2$ [18]. Presently, all these model are at the evaluation phase, which means that the upgradation of datasets is necessary. The ability to represent the results in a concise manner becomes crucial in order to provide the elements that are useful for decision-making purposes.

Except for the model developed by Chahoud et al. [18] to study the feasibility of an artificial recharge structure, the southern part of the delta has been explored less regarding hydrogeological characterization and groundwater model studies. While a few studies [19–22] have focused on the northern part of the Po delta, so far, no hydrogeological study has modelled the complex multi-aquifer system of the first three aquifers of the southern part of it (southern of Ravenna), and this area remains partly unexplored from the hydrogeological modelling point of view.

The present paper develops for the first time a three-dimensional multilayer numerical model for the southern coastal part of the southern Po delta by means of conceptualization of the multilayered aquifer and calibration of the spatially distributed elevation of the groundwater table and piezometric head from the three-dimensional aquifers. Modflow is utilized in the development of the regional scale conceptual and numerical groundwater flow model. The developed model was designed to investigate the present status of aquifer exploitation by groundwater pumping and artificial drains. The results of this study are preparatory for the implementation of an integrated density-dependent hydrogeological model to assess the salinization of the aquifers by means of the seawater intrusion phenomenon together with other challenges such as land subsidence and the unsustainable use of water during the summer period. As outlined by Giambastiani et al. (2021) [23], the Emilia Romagna phreatic coastal aquifer is affected by salinization, which reduces freshwater availability. Moreover, groundwater salinization is promoted by land subsidence. Antonellini et al. [24] mention the increasing scarcity of freshwater resources in the southern Po delta because of intense use, salinization and long periods of drought. They also refer to many abusive wells employed for the tourist establishments on the beach. According to the Po River Basin Authority, from press reports, in fact, it emerges that the Po River Basin Authority estimates the number of existing wells in Italy at as many as ten million, including registered, illegal, active and abandoned wells [25].

In this context, already in the late 1990's, Farina et al. (1988) [26] roughly estimated the number of abandoned wells to be 1 per km$^2$ in the Emilia Romagna region; they also evidenced the potential risk, as they could constitute a preferential pathway for cross-contamination between aquifers.

The model is conceptualized using a litho-stratigraphic model developed using the available litho cross-section profiles, estimated river flow and stage, drains, measurements of groundwater table and piezometric head from multilayer aquifers obtained from Agenzia Prevenzione Ambiente Energia Emilia-Romagna. There are 21 wells and piezometers that

penetrate a shallow unconfined aquifer and two deep-seated confined aquifers used in the study. Each well and piezometer selected for this study is equipped monofilters to replicate the hydrogeological condition that exist in the specific aquifers.

## 2. Materials and Methods

### 2.1. Study Area

The study area has a surface of 537 km$^2$, is located at the UTM zone 32° N and ranges between 752,260 m to 791,561 m E and 4,876,892 m and 4,917,405 m N, which is the southern part of Po delta (Figure 1).

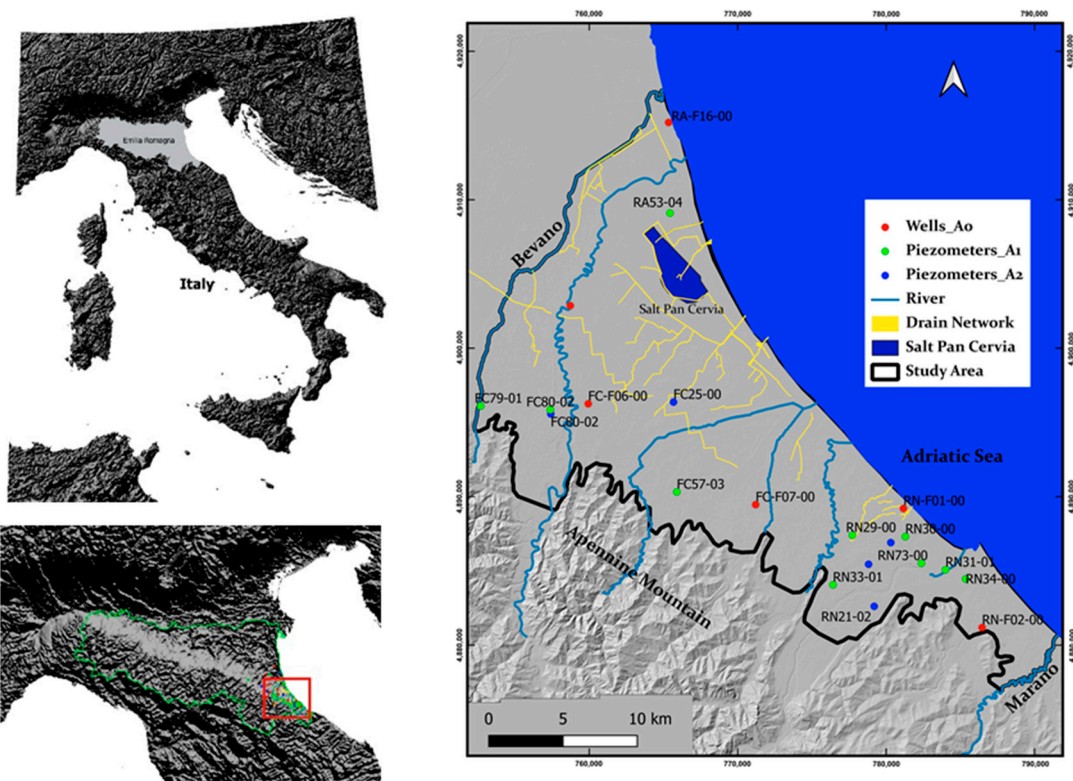

**Figure 1.** Base map of the study area.

This area is not free of threats such as storm, seawater intrusion, soil erosion, shore line alteration and especially land subsidence caused by groundwater pumping and soil compaction [27–29]. The Apennine Mountain chain, together with the Alps Mountain chain, create an orographic barrier to the cyclones formed in the Atlantic Ocean and in the area near the Iberian Peninsula [30]. By reducing the moist air circulation in this way, the rainfall is minimized at the leeward side, the side which covers the present study area. Similarly, in the study area, the Adriatic Sea has two different wind directions, namely Scirocco, more frequent with low energy from SE, and Bora, less frequent with high energy that brings more rainfall. The resultant average rainfall from the combination of all actions is 676 mm/yr estimated between the years 2001 and 2012. The average temperature is 14.8 °C and categorized as a temperate area on the basis of the Koppen climate classification [31].

### 2.2. Geomorphology of the Study Area

In the study area, the average land subsidence of 5 mm/yr, which ranges between nil and a maximum of 10 mm/yr, is reported in the southern part [14,16,32]. The impact of natural calamities is believed to be less in the north because of the presence of coastal dunes; even though they are discontinuous and sparse, they function as flood protection dykes and seasonal embankments. The eastern side is bounded by the Adriatic Sea with

a localized treat and retreat of shore lines [33,34]. These scattered dune surfaces help the formation of freshwater lenses that could stand against seawater intrusion; however, the majority of coastal land experiences a polder situation [35]. The topography is smooth and flat with a slightly elevated coastal dune surface, which is steeply elevated at the Western part. The elevation ranges from $-3.40$ m above the mean sea level to 105 m above the mean sea level. The western part is bordered by the foothills of the Apennines Mountains and characterized by a recent deltaic formation on the local scale on the top. These recent deltaic formations in the form of localized alluvial fans are followed by old and oldest alluvial fans' formations, and they extend to various depths and spaces with a layer effect [36]. The N and S boundaries of the study area have been chosen as the rivers Bevano and Marano to maintain natural hydraulic boundaries only at the shallow aquifer. The artificial drainage network, on the other hand, is densely present in the NE part because the land surface is below the mean sea level. This area forms the part of land reclamation monitored by "Consorzi di Bonifica della Romagna". The area having a lower elevation of $-3.40$ m above the mean sea level serves as saltpan (Cervia) sourced by seawater through a canal and by water drained artificially from the shallow aquifer. The artificial drains help to maintain the groundwater table at $-0.5$ m above mean sea level and the water is pumped into the channels for agricultural activities and for disposal into the sea [22]. On the other hand, the artificial drains encourage a negative hydraulic gradient from the coastal boundary. This also reduces the existence of freshwater by draining it into the channel and by which there is an upwelling and mixing of deep-seated connate water in to the freshwater zone above [35].

The water derived from the atmosphere to earth surface above the rate of infiltration at the particular ground surface results in a runoff and the formation of surface water bodies with their associated sedimentary landforms, such as the delta in the study area [37]. The time required for the circulation of water in this way between atmosphere and surface water bodies is several times less than the time required for water circulation between the atmosphere and groundwater [38]. Any change in their primary driving forces, such as evaporation, transpiration, a combination of these two, evapotranspiration through the plants, and human intervention, caused alterations in surface water bodies, land forms and their associated activities. The global volume of evaporation loss in the stagnant water bodies, such as lake and dams, are reported as $1500 \pm 150$ km$^3$ yr$^{-1}$ with an increment of $3.12$ km$^3$ yr$^{-1}$ [39]. Due to human intervention, the construction of dams is changing the land use pattern, which leads to alterations of landforms in the downstream areas [40,41]. Additionally, the surface water is very much utilized because of immediate access and nil or less requirement for treatment. Hence, the surface water is shrinking and being contaminated because of anthropic interventions. On the other hand, the shrinking of the surface runoff in the flat terrain is highly sensitive to changing their path and causing a breakage in the flow and sediment loading.

Apart from the Po River, some other rivers and streams, function as tributaries and distributaries of Po, and many other perennial and non-perennial rivers and streams are used for agricultural activities and draining water into the Adriatic Sea. Likely, the ground surface of the present area is served and drained naturally into the Adriatic Sea by six non-perennial fragmented rivers via estuaries and river outlets. There is one stream called Pisciatello contributing to Rubicone at the rate of $0.29$ m$^3$s$^{-1}$ to $1.97$ m$^3$s$^{-1}$ to the River Rubicone (Agenzia Prevenzione Ambiente Energia Emilia-Romagna). The coastal zone is mostly flat, highly fragmented and altered by natural and manmade structures, such as rivers, artificial drains, roads, etc. The geomorphological features are the alluvial fan, alluvial plain and coastal plain by the action of these rivers, stream and marine sedimentation.

The significant anomaly in the rainfall pattern during the 1980's caused non-equilibrium at the mouth of the river [33]. Hence, in addition to tectonic activity and land subsidence, the coastal morphology is being modified by the volume of surface runoff. Accordingly, the reduced volume of flow in the river is reducing sediment loading and the deposition

of sediments in the downstream and at the mouth of the river by which the meandering, shifting, widening and deepening of the river mouth occurs against the action of tidal waves (0.8 m). The volume of the river flow, tidal forces, and geometry of the river mouth and estuarine geometry are the parameters controlling active sea water intrusion in these typical low-lying areas [42].

Investigating the river flow volume, critical depth, elevation and slope of the river bed and the elevation of the river water level at closed conditions helps in the identification of the distance of the direct entrance of seawater into the river channel. The estimated distance in the river channel from the coastal boundary using the above parameters is considered vulnerable to seawater intrusion. In the model development, the river channel identified as vulnerable is assigned with seawater properties.

The topography, geometry of the river bed and basin boundaries of the study area were processed and analyzed using a digital elevation model at the resolution of $5 \times 5$ m, provided by Agenzia Prevenzione Ambiente Energia Emilia-Romagna. The basin boundary was analyzed by Sthraler stream order classification [43]. The working principle of the Sthraler stream order is based on connecting the channel that falls under the same order. The basin boundary used in this study is estimated using the fifth order.

### 2.3. Litho-Stratigraphic Units and Hydrogeology of the Area

The subsurface is made up of multilayer Quaternary sediments of marine origin at the bottom, of fluvial origin at the top and it is composed of gravel, sand, silt and clay [44]. It is formed by the tectonically active Alps and Apennine Mountain chains. The raising and glaciation of the Alps and Apennines have progressively moved the bordering Adriatic Sea away from the ancient Po Gulf. The space left by the transgression of the sea is filled with deltaic and shore face sediments during the seven phases of the geological periods, between the Pliocene era until recently [36]. This sedimentation process may vary according to the geological setting, climatic condition and their related alteration in the shore line and elevation of sea level. Climate and tectonic activities have formed a sequence of continuous and discontinuous patches of lithological layers spreading laterally at varying sizes, ranging from a few meters into the overall basin. The accumulation, sorting, layering and compaction of each layer has occurred under a controlled environment over several geological time periods. The fine-grained sediments, clay and silt, represent either a low-energy depositional environment when the flow volume in the river and stream is low or a period of marine transgression. The coarse-grained sediments, sand and gravel, represent either high-energy deposits during a period of high-volume flow in the river and stream or a period of marine regression. The treat and retreat of the Adriatic Sea shore in the seven phases, quantified between up to a distance of 30 km towards W and 250 km towards SE from the present shoreline, encounter connate water trapping and organically enriched clay sediments between the deltaic layers [20,45]. In recent decades, the process of sedimentation is reduced and enough only for the formation of small-scale alluvial fans [46]. According to the present geological settings and climatic condition, 500 to 600 tons of sediment deposits are estimated per sq.km at one of the river basins draining the Apennine Mountain, which hardly reaches the margin of the sea [47].

### 2.4. Hydrogeology of the Study Area

The overall thickness of sedimentary formations forms three hydrogeological groups, namely A, B and C [48–50]. The thickness of Groups A, B and C increases by up to a depth of 200 m, 350 m and 2000 m below ground level (bgl) (Figure 2) [18]. The grouping is carried out not on the basis of the lithological type and similarity in formations. It is carried out on the basis of generic information such as the seven phases of treat and retreat of Adriatic sea, the period of non-deposition, the same periods of deposition and, more evidently, the changes in the texture of grains and their depositional mechanisms. These groups of deposits, namely Groups A, B and C, are again classified into distinctive hydrogeological units for the same reason along with impermeable bounds caused by climatic oscillations,

the treat and retreat of sea, and tectonic oscillations within the same group. Each unit is called "Hydrostratigraphic Sequential Unit", which behaves as a separate aquifer [49]. The first three aquifers of Generic Group A (A0, A1, A2 and $A_{base}$) are addressed in the present study. In some cases, when $A_{base}$ is not included in this study, it is further classified into A3 and A4. Laterally, the aquifers are classified into: (i) Free aquifers formed by alluvial fan at the foot hills of the Mountain; (ii) Upper confined aquifers composed of A1 and A2; (iii) Lower confined aquifers comprising an A3 and the following layers; and (iv) Plain aquifers along the coastal part. A0 aquifer alone is classified into: (i) A phreatic aquifer formed by the river and (ii) A phreatic aquifer formed by coastal sediments. The shallow aquifer is recharged by rainfall and river channels, whereas the deeper aquifers are recharged remotely in the zone of free aquifers formed by alluvial fans.

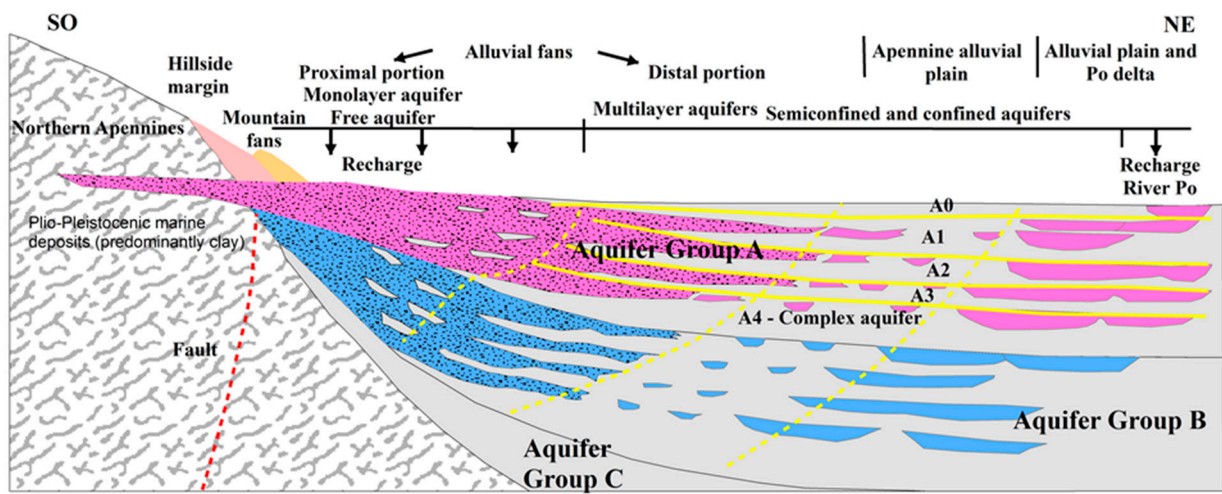

**Figure 2.** Hydrogeological setting of the Po delta.

*2.5. Numerical Model Development*

MODFLOW 2000 [51], a finite difference numerical code is used in this study to simulate the groundwater system under constant density assumption. Generally, the saturated groundwater flow field is governed by the following partial differential equation:

$$\nabla \cdot (\mathbf{K} \cdot \nabla h) - W = S_s \frac{\partial h}{\partial t} \tag{1}$$

where $h$ (L) is the potentiometric head, $\mathbf{K}$ ($LT^{-1}$) is the hydraulic conductivity tensor, $W$ is volumetric flux per unit value representing sources and/or sinks of flux ($W < 0.0$ for outflow of the groundwater system, $W > 0.0$ for inflow ($T^{-1}$), $S_s$ = specific storage of saturated porous material ($L^{-1}$), and $t$ = time.

MODFLOW 2000 solves Equation (1) by means of finite difference scheme. A three-dimensional structured grid has been created according to the geo-lithological setting. Then, the vertical and horizontal conductance terms between the cells of the finite difference grid are determined according to their direction, which can be normal or parallel to the bedding.

The model accuracy was evaluated using standard error statistics, such as coefficient of determination R-squared ($R^2$), mean error (ME) (Equation (2)), root-mean-squared error (RMSE) (Equation (3)) and model efficiency (EF) (Equation (4)).

$$ME = \frac{1}{N} \sum_{i=1}^{N} O_i - P_i \tag{2}$$

$$RMSE = \sqrt{\frac{1}{N} \sum_{i=0}^{N} (O_i - P_i)^2} \tag{3}$$

$$\text{EF} = 1 - \frac{\sum_{i=0}^{N}(O_i - P_i)^2}{\sum_{i=0}^{N}(O_i - \overline{O})^2} \tag{4}$$

where $N$ is the number of wells/piezometers, $O_i$ is the observed groundwater table or piezometric head, $P_i$ is the simulated groundwater table or piezometric head and $\overline{O}$ is the mean observed groundwater table or piezometric head.

## 3. Results

### 3.1. Lithology and Hydrogeological Characterization

A subsurface geolithological model of selective aquifers was developed using seventeen cumulative lithological cross-sections with the horizontal resolution of 1: 10,000 collected from the seismic and soil geological service agency, Servizio geologico sismico e dei suoli (Figure 3a). The term cumulative is used to represent the cross-sections that are derived using a wide range of sources, such as seismic data, electrical logs, borehole logs and penetrometers. The horizontal and vertical distribution of each formation is originates from the action of rivers and streams running from the Apennine Mountain and sea, and their horizons were interpreted using 74 horizon IDs by making use of sixteen cross-sections perpendicular to the sea and one cross-section more or less parallel to the sea (Figure 3a,b).

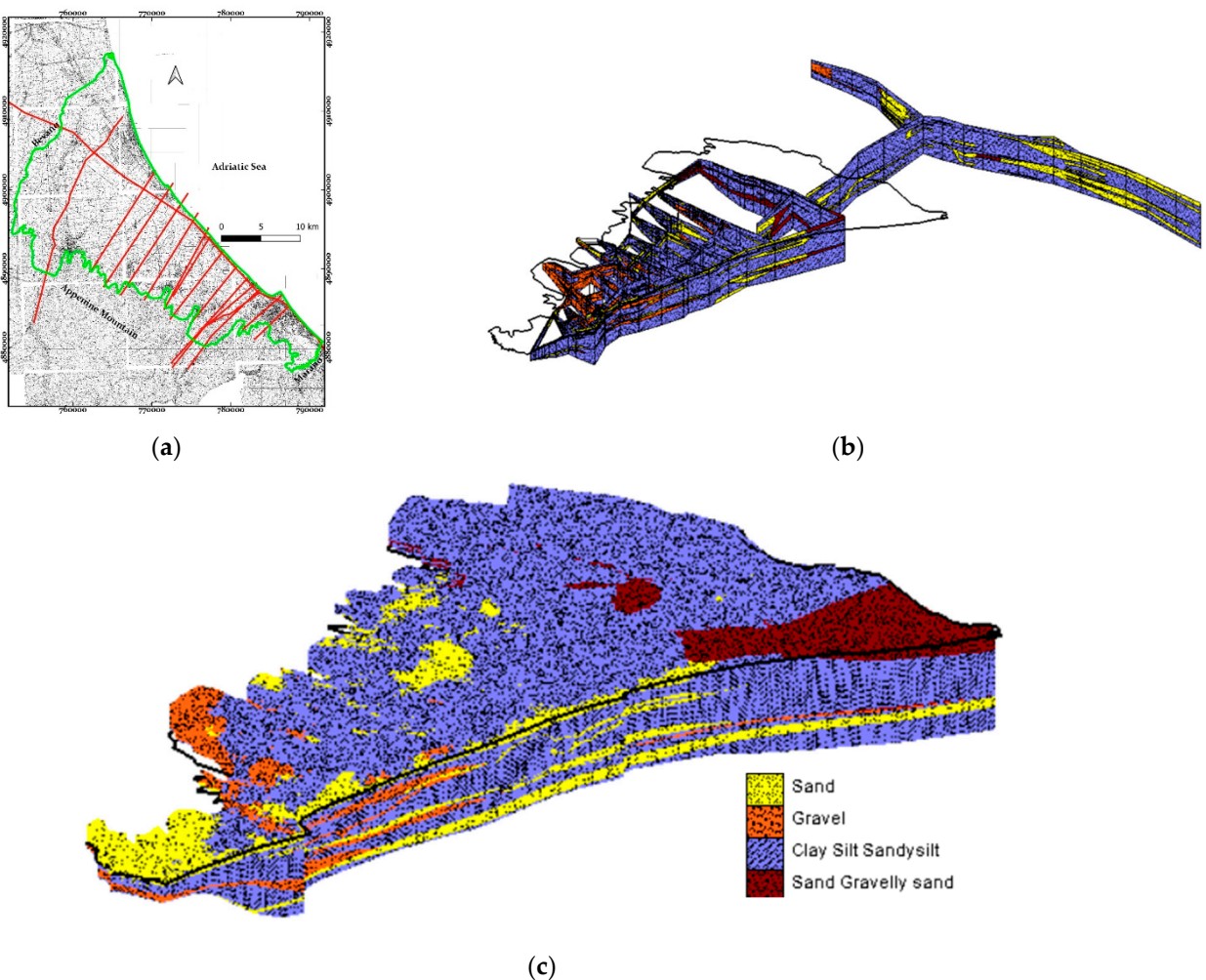

**Figure 3.** (**a**) Location of cross-section profiles; (**b**) Construction of litho-stratigraphic model using horizon of cross-section profiles; (**c**) Subsurface litho-stratigraphic model of the study area.

The groundwater data measured at sparse locations show that the piezometric head was at few meters above the ground surface [52]. Subsequently, it went to 30 m below the actual value due to heavily uncontrolled exploitation of groundwater from all the shallow and deep-seated aquifers [14]. The recovery took place from the late 1970's after the construction of water outsourcing structures, and the groundwater table and quality are monitored at an increased number of locations. At present, the spatial and temporal variation of the groundwater system in the shallow and deep-seated aquifers are monitored using 740 wells and piezometers, in which 21 wells and piezometers with reasonable data history fall inside our study area, i.e., 6 wells from A0 aquifer, 10 wells from A1 aquifer and 5 wells from A2 aquifer. The depth of penetration varies between 2.14 m and 21 m in the A0 aquifer, 25 m and 120 m in the A1 aquifer and 27 m and 214 m in the A2 aquifer. The wells and piezometers selected for the study are facilitated with monofilters to target their respective aquifer. There are a few more wells that are not considered in this study due to the absence of and irregularity in the data availability. The elevation of the groundwater table varies from −0.28 to 24.46 m in A0 aquifer (Figure 4a,b), with the maximum fluctuation of 3.3 m in the well located in the norther central part. The piezometric head is between −11.43 to 58.73 m above mean sea level in A1 and −2.16 to 19.27 m above mean sea level in A2 aquifer.

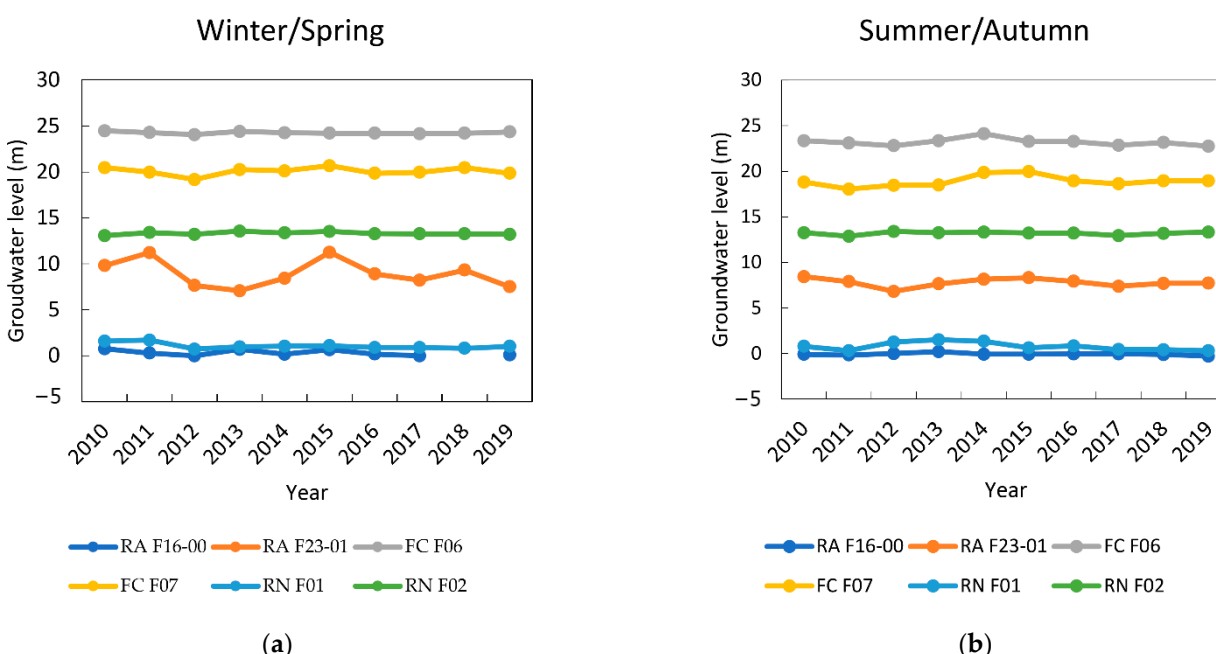

(**a**)                                                                           (**b**)

**Figure 4.** Elevation of groundwater table during winter (**a**) and summer (**b**).

### 3.2. Surface Water Simulation

The monthly average volume of river flow measured at the monitoring stations located at the distance of 2.39 km to 48.6 km from the mouth of the river varies from 0.06 m$^3$s$^{-1}$ in the Rubicone River (prior the contribution from the Pisciatello Stream) to 33.00 m$^3$s$^{-1}$ in the Marecchia River (Figure 5). The basins covering the north and south boundaries of the study area, named the Bevano Basin and Marano basin, are lacking a monitoring station and long time series of data. The hydraulic simulations were carried out for all the major basins for a period of 3070 days using HEC-RAS 6.2. developed by U.S. Corps of Engineers [53]. The simulation also includes a stream, namely the Pisciatello Stream, a tributary of the Rubicone River, which enters the main river channel a few kilometers away from the river mouth.

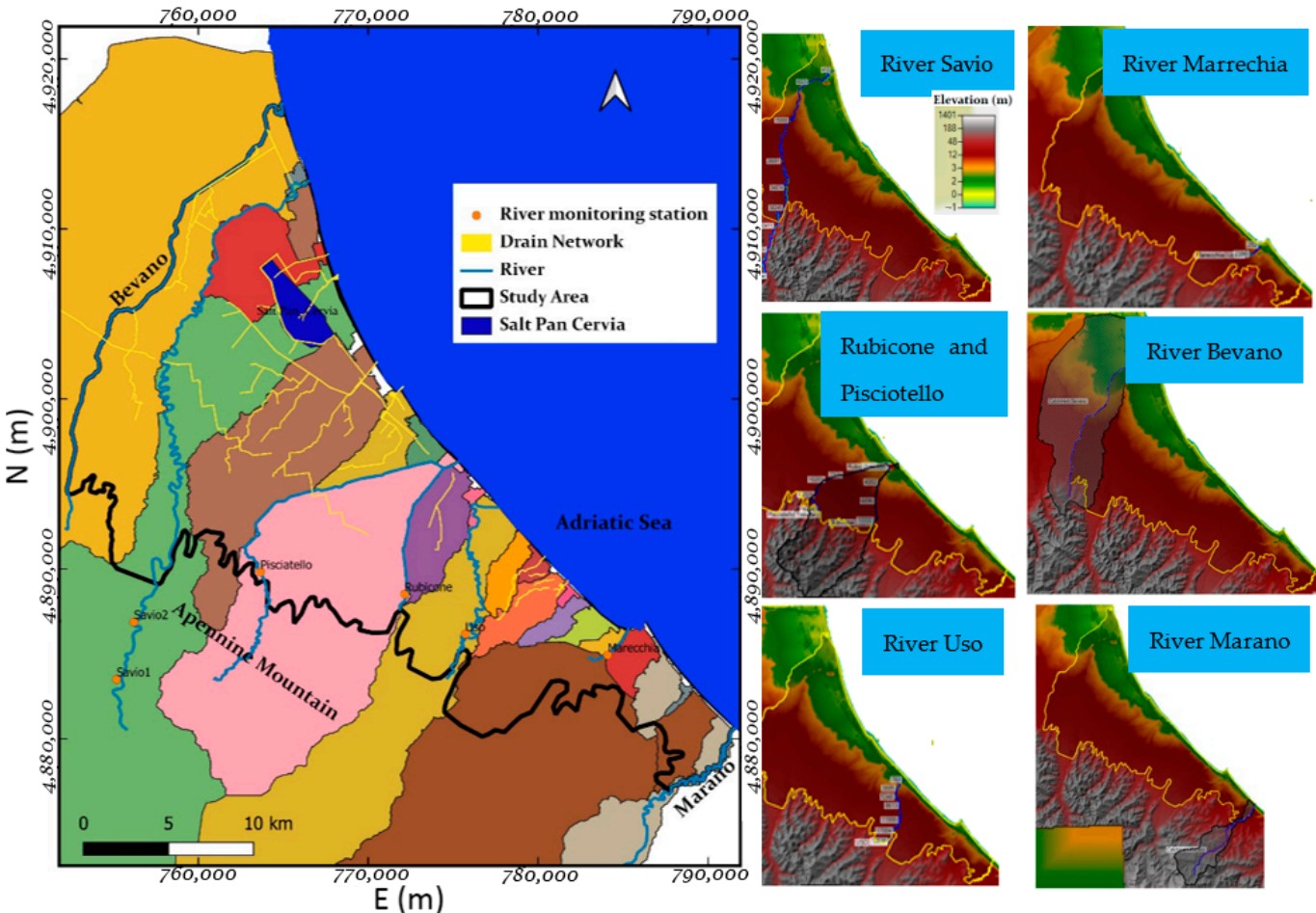

**Figure 5.** Location of river water monitoring stations, basin boundary obtained by strahler order 5 and hydraulic simulations of each basin.

The measured monthly flow rate at the stream is 0.29 m$^3$s$^{-1}$ to 1.97 m$^3$s$^{-1}$. For the two basins that are covering the north and the south of the study area, the simulations were conducted using average rainfall values due to the lack of monitoring stations. The Manning value representing the roughness coefficient of the river channel is assigned on average of 0.05 at the middle of the river channel and 0.035 at the river banks. The flow volume for the basins lacking monitoring stations, the elevation of river water stage and the critical depth at several locations along the river channel have been estimated under a closed condition. The term closed condition indicates that a boundary condition neglects the influence of the sea. It ensures that the simulated parameters in the cross-section are based on the hydraulic condition that is present in the upstream. In other words, the cross-sections located at the mouth of the rivers do not represent the impact of seawater. In this way, the estimation of parameters will be helpful in understanding the behavior of the river alone at the existing basin characteristics.

The data processed by the Strahler order above 5 show 6 major and 19 micro basins (Figure 5). It is evident from the presence of micro basins along the coastal boundaries that the coastal morphology is unstable and it is also disturbed by manmade activities. Due to flatness, small morphometric features such as dunes, dyke, roads and levees become micro basins. The major basins have an area between 62.5 sq.km and 585.5 sq.km, with the topographic elevation varying from 1405 m above mean sea level at the Apennine catchment and −1.03 m above mean sea level at the river mouth. The area with the least elevation of −3.40 m falls in the localized minor basin due to the lower ground altitude with artificial channels and direct connectivity to seawater. All the river channels are protected by the embankments' and dykes' construction at various distances from the sea. The flow

volume obtained at the mouth of the river using monthly average flow volumes measured at the upstream varied from 0.11 m$^3$s$^{-1}$ in the Savio Basin to 112.31 m$^3$s$^{-1}$ in the Bevano Basin (Table 1).

**Table 1.** The data measured at the upstream (Agenzia Prevenzione Ambiente Energia Emilia-Romagna) and the data extracted from the digital elevation model and hydraulic simulations using HEC-RAS.

| Name of Basin | Area (km$^2$) | Max Flow (m$^3$s$^{-1}$) | Min Flow (m$^3$s$^{-1}$) | Average Flow (m$^3$s$^{-1}$) | Distance to 0 m River Bed Elevation (m) | Slope | Distance of Monitoring Station to the Sea (km) | Highest Elevation of Basin (m) | Least Elevation of Basin (m) | Estimated Flow at the Cross-Section (m$^3$s$^{-1}$) | Elevation of Water Stage (m) | Distance to Cross-Section from the Sea (m) |
|---|---|---|---|---|---|---|---|---|---|---|---|---|
| Savio | 585.8 | 574 | 0 | 8.16 | 2326 | 0.0004 | 48.60 | 1361 | −1.03 | 0.11–22.8 | −0.94 to −0.44 | 413 |
| Pisciatello and Rubicone | 172.1 | 106 | 0 | 0.33 | 763 | 0.001 | 10.07 | 469 | −1.02 | 0.46–2.09 | 0.71 to 0.75 | 86 |
| Uso | 109 | 338 | 0 | 0.77 | 337 | 0.0009 | 19.62 | 762 | −0.32 | 0.15–2.57 | 0.71 to 0.78 | 228 |
| Bevano | 334.7 | - | - | - | 1790 | 0.0005 | - | 170 | −0.99 | 8.49–112.31 | 0.04 to 1.3 | 0 |
| Marano | 62.4 | - | - | - | 123 | 0.003 | - | - | −0.4 | 0.3–13.65 | 2.71 to 3.10 | 0 |
| Marecchia | 531.8 | 636 | 0 | - | 407 | 0.002 | 2.39 | 1405 | −1.0 | 1.39–32.99 | −0.88 to −0.33 | 282 |

### 3.3. Model Conceptualization

The model domain covers an area of 537 km$^2$, with a maximum distance of 45 km in the NW-SE direction, and of 26 km in NE-SW direction. After an alignment along the principal axis of the flow direction between the Apennine Mountain and the Adriatic Sea, the model domain is laterally discretized by 192 rows and 118 columns (Figure 6). There are 8683 active finite difference grids with the size of 500 m$^2$ covering the targeted area of the numerical model simulation. The size of the grid is approached on the basis of several attempts in order to increase computational efficiency and without compromising the representation of the geological conditions existing in the study area, such as the dominance of flat topography, groundwater table, regular boundary between the aquifers, etc.

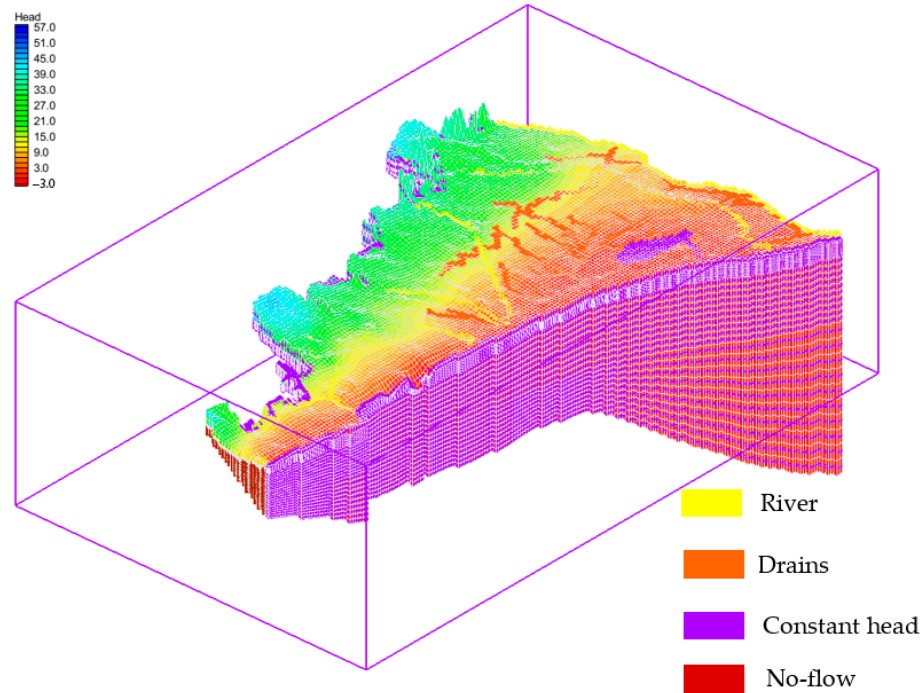

**Figure 6.** Conceptualization of the model domain.

The top surface of the model domain varying from −1.19 m to 105 m above the mean sea level is imported from the digital elevation model at the resolution of 5 × 5 m, provided

by Agenzia Prevenzione Ambiente Energia Emilia-Romagna. Since the data points from the digital elevation model are located at a distance of 5 m each, the elevation of each grid in the model is an average value obtained from the group of data points falling inside.

The vertical thickness of the model is conceptualized up to a depth of $-330$ m below the mean sea level, which forms the base of the A2 aquifer based on the litho-stratigrapic model (Figure 3c), developed using cross-sectional profiles. Due to the occurrence of discontinuous patches of less/nil permeable units and the connectivity between the permeable units, the 'Hydrostratigraphic Sequential Unit' boundary between the layers [49] is fixed as the aquifer boundary. The model domain is vertically configured into a total of 30 layers to represent each aquifer, A0, A1 and A2, by 10 layers. The respective thickness of each layer varies from 0.2 m along the Apennine Mountain to 10 m and 21.8 m along the NE part of the A1 and A2 aquifers. In the A0 aquifer, the thickness of each layer is configured on the basis of the saturated thickness of the aquifer to avoid non-convergence between the nodes.

The boundary condition ensures the relationship that exists between the model area and the surrounding environment. Assigning a proper boundary condition demands a detailed investigation of the hydrogeological behavior of each aquifer, which is influenced by many factors. Since the model is a multilayer aquifer system, the boundary condition has strong implications on the geomorphological features of the shallow aquifer. The north and south of the model domain are defined by the Bevano and Marano Rivers, respectively. The presence of an aquitard at the base of the A0 aquifer and coincidence of both the historical [14] and recently measured negative elevation of the piezometric head at the deep-seated A1 and A2 aquifers confirms the existence of a diverse hydrogeological interaction between A0 and deep-seated A1 and A2 aquifers in the northern part. Towards the south, the aquitard demarcates the base of A0 and the elevation of the piezometric head in both the A1 and A2 aquifers, which are observed to be above the mean sea level. The gradient of the piezometric head is parallel to the southern boundary. The river boundary condition is assigned along the north and south boundaries of the shallow A0 aquifer. The parallelism of the piezometric gradient downward into the sea at the A1 and A2 aquifers along the southern boundary indicates the nil or negligible exchange of mass between the sides of the boundary. Hence, the no-flow boundary is assigned along the southern boundary of both deep-seated aquifers. Towards the north, the piezometric gradient is perpendicular to the boundary representing a negative outward flux from the model domain. However, the historical elevation of the piezometric head remains stable all the time. Hence, a constant head boundary using an elevation of piezometric heads measured at the nearby piezometers is assigned in the deep-seated aquifers. The Adriatic Sea boundary on all the layers and salt pan limited only to a shallow A0 aquifer is represented by a constant boundary condition with a mean sea level of 0. The western boundary represents the foot hills of the Apennine Mountain characterized by alluvial fans considered as recharge windows for all the aquifers. The constant head boundary condition is assigned by using a representative elevation of the groundwater table and piezometric head at the nearby wells and piezometers.

The impacts of the river, artificial drainage and navigation canals are encountered using the RIV1 and DRN1 packages provided in the GMS (groundwater modeling system) 10.4v software program. The geometry of surface water channels and drains is significantly important in the coastal zones at polder environments. The grids were refined along the path of surface water channels and drains that enable the repair of the river bed elevation extracted from the digital elevation model at the interval of 500 m along the x-axis (i horizontal plane). The average elevation of water stage extracted from the cross-section during the month of January is assigned. The drains are purposefully maintained to keep the groundwater table and surface water stage at $-0.5$ m below mean sea level. Considering the density of drains at the low-land topography, the average drain-bed elevation is assigned at 0.5 m below the surface grids. The values of the conductance of the river bed and drain bed are assigned as 0.009 m$^2$d$^{-1}$ and 2.5 m$^2$d$^{-1}$, which are used in the northern part of the delta, which has the same soil texture and hydrogeological setup [49].

*3.4. Model Calibration*

The model calibration is the process in which the accuracy between the observed and simulated values of the groundwater table and piezometric head are targeted by adjusting the appropriate aquifer parameters. Initially, calibration is achieved in a steady-state condition under the absence of the time variable source and sink factors. In a transient state condition, calibration involves time variable sources and sinks that only follow after an estimation of the aquifer parameters by achieving a good match between observed and simulated values in a steady-state condition. The aquifer parameters vary with respect to space and depth. The porosity obtained by conducting a grain size analysis of coastal sediments varies from 0.19 to 0.76, with an average porosity value of 0.3 [50] and an effective porosity value of 0.25 [49]. The hydraulic conductivity obtained by the grain size analysis using Hazen's formula, pumping test, permeability test, geoelectrical resistivity survey using Archie's law, slug test and tidal well test varies between 0.0003 md$^{-1}$ and 86.4 md$^{-1}$ [19,22,35,54,55]. The groundwater table and piezometric head measured during winter in the year 2010 is assigned as the initial head. The bottom layer of each aquifer is assigned with a hydraulic conductivity value of 0.001 md$^{-1}$, with a vertical anisotropy value of 0.1, to represent the existence of an aquitard. To avoid the computational delay, the total thickness of HGU is calibrated individually for horizontal hydraulic conductivity by inverse modelling, using the equally distributed pilot points option provided by PEST [56]. The estimation of hydraulic conductivity has been carried out separately for each aquifer, and the interactions between the aquifers in terms of the elevation of the groundwater table, the piezometric head difference, and the boundary conditions have been neglected. These reference values are introduced in the multilayer model and calibrated manually using trial-and-error approaches. The aquifer parameters obtained from the steady-state calibration are shown in Table 2. The accuracy of the model calibration is obtained with the RMSE values of 1.1, 3.3 and 5.0 in A0, A1 and A2 aquifers (Figure 7).

**Table 2.** Calibrated aquifer parameters.

| Parameter | Value | Unit |
|---|---|---|
| Horizontal hydraulic conductivity | A0: $5.79 \times 10^{-5}$ to $9.26 \times 10^{-4}$<br>A1: $1.16 \times 10^{-5}$ to $2.45 \times 10^{-3}$<br>A2: $1.16 \times 10^{-6}$ to $4.70 \times 10^{-3}$ | ms$^{-1}$ |
| Vertical anisotropy | 1 | (-) |
| Effective porosity | 0.3 | (-) |

It is noticed that the coefficient of determination, $R^2$, is significantly above 0.87 in all the three aquifers (Table 3), which means that the matching is achieved between the observed and simulated groundwater table in the shallow A0 aquifer and the piezometric head in the deep-seated A1 and A2 confined aquifers. The mean error and root-mean-squared error values vary between 1.11 in the shallow and 5.07 in the deeper aquifer. The model efficiency values are above 0.50, which means that the model is calibrated with a good agreement between observed and simulated results. Thus, the developed multilayer model under steady-state conditions is fit for further transient simulations.

**Table 3.** Accuracy of the model calibration.

| Parameter | A0 Aquifer | A1 Aquifer | A2 Aquifer |
|---|---|---|---|
| Mean error | 0.71 | 1.37 | 3.25 |
| Root-mean-square error | 1.11 | 3.33 | 5.04 |
| EF | 0.98 | 0.89 | 0.73 |
| R | 0.99 | 0.94 | 0.87 |

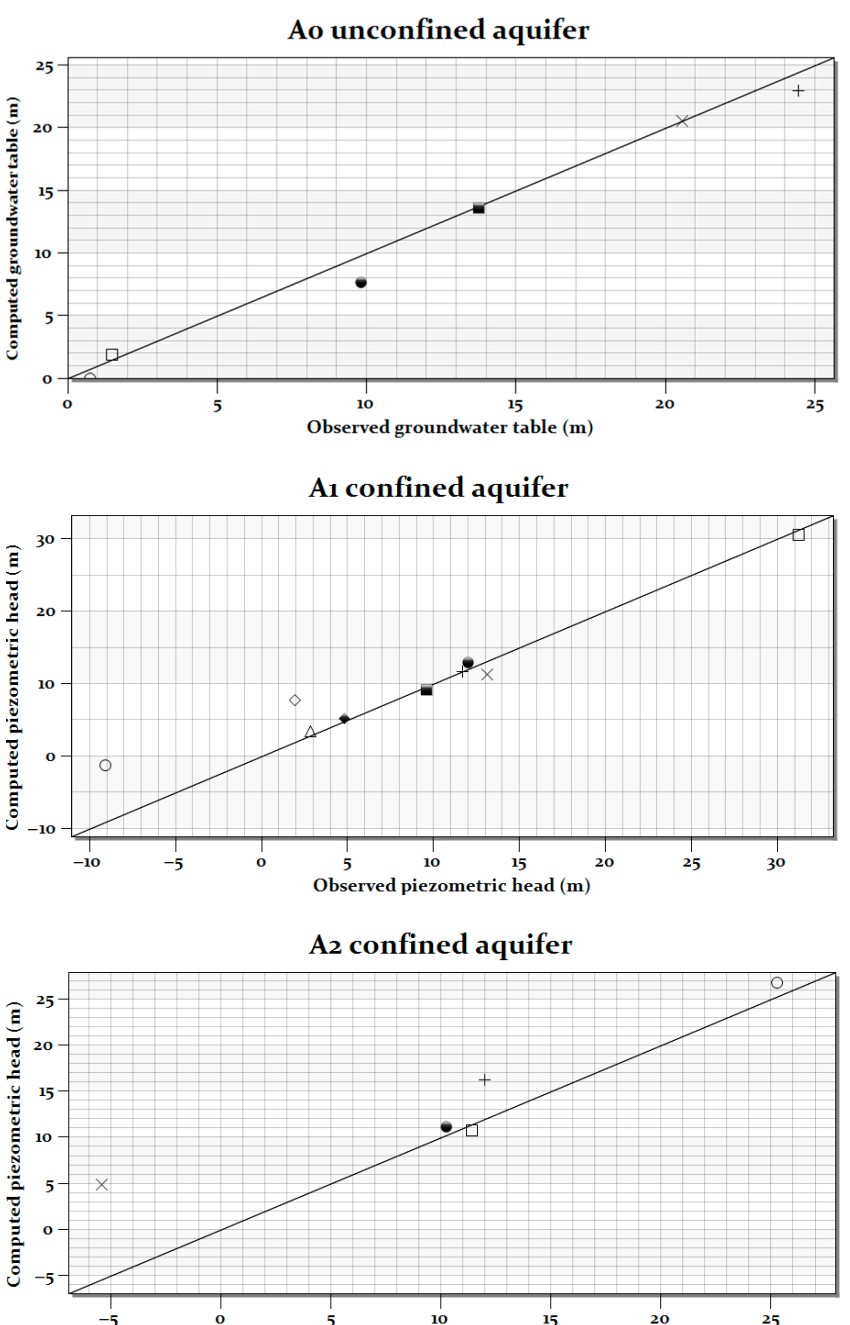

**Figure 7.** Calibration of the multilayer aquifer under steady-state condition.

## 4. Discussion

Volumetrically, permeable formations are high in the confined aquifers, i.e., A2 followed by A1, with numerous low conductive intercalations and less so in A0 (Figure 8). The ground surface located between the coastal plain and alluvial fan area is an alluvial plain composed of fine-grained sediments (clay, silt and sandy silt) by the action of rivers and streams. Recent investigations prove that the flow volume in all the rivers has decreased because of drought and the finer sediment accumulation, which allows a minimal rate of infiltration. Below the fine-grained soil, the sporadic extension of sand from the alluvial fan and coastal sand forms the unconfined aquifer, as detailed by Amorosi et al. [45]. The A0 aquifer is under the development process that started from the period of late Pleistocene. From the cross-section profiles, it is noticed that the bottom of A0 at the coast is governed by organic clay deposited in a marsh and swampy environment [20]. The permeable for-

mations, such as gravel, are high in the SW part (alluvial fan) and sand is high in both SW and the coast plain. At some places, alluvial fans mediate the shallow unconfined aquifer to confined A1 and A2 aquifers at the bottom. This connectivity is also noticed in the NE low-lying area, as reported by Chahoud et al. [18], which is made up of a mixture of sand and gravelly sand.

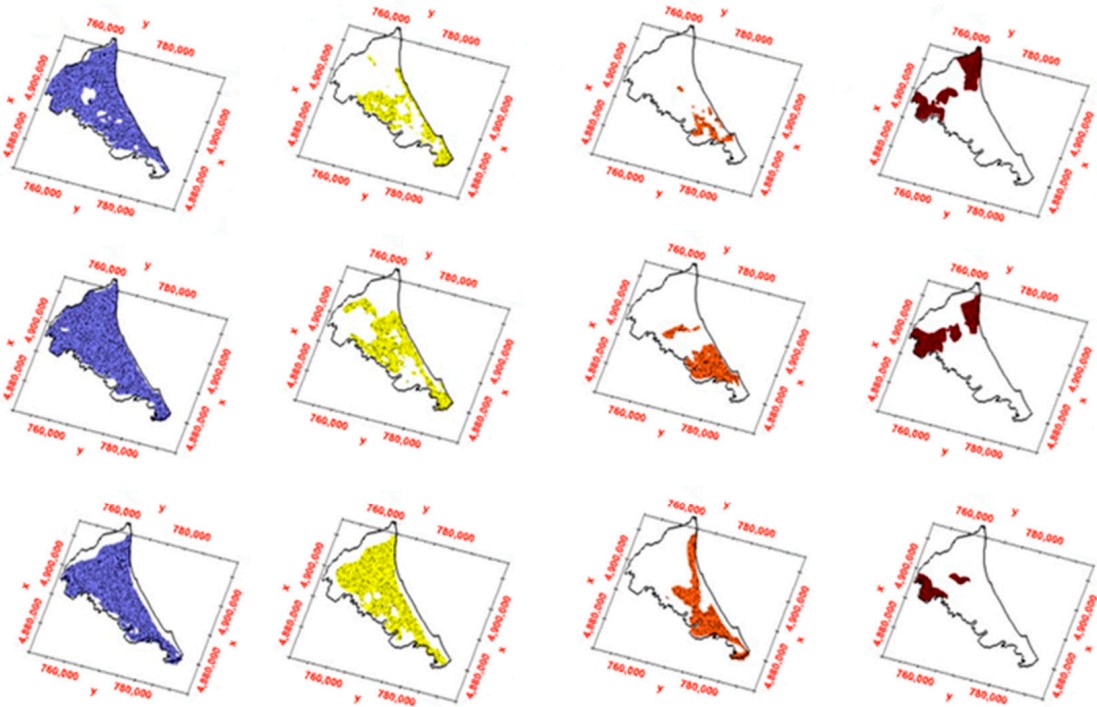

**Figure 8.** Volumetric comparison of four types of sedimentary formations at each HGU (aquifers, A0, A1 and A2).

The three-dimensional spatial distribution of the groundwater table and piezometric head (Figure 9) indicates that all the aquifers have a common apical part along the W and SW (Apennine margin). The groundwater flow occurs from the Apennine alluvial fans to the Adriatic Sea in all the aquifers with remarkable changes in the NE part. The NE part in the A0 unconfined aquifer is a low-lying area in which the flow is slightly localized and turned below the mean sea level by artificial drains. In the deeper aquifers, the flow direction at the NE part is severely impacted by groundwater pumping from the deep-seated wells. The regional cone of depression in the industrial area (Ravenna), located just northward of the study area, shrinks the potentiometric surface in the deep-seated aquifers that drive the flow northward instead of eastward into Adriatic Sea in both the A1 and A2 aquifers. There is a connection between the A1 and A2 aquifers in the NE part of the area.

The river bed elevation extracted from the digital elevation model is negative at the mouth of all the rivers. For the given volume of flow, the critical depth is also negative in all the rivers except the Uso River. Hence, the occurrence of seawater entering through the river mouth of all the rivers may exist. The river bed elevation of 0 m, an imaginary equipotential zone where the influence of seawater is assumed to be lower, is located at a distance of 2326 m followed by 1790 m inland from the coastal boundary of the Savio and Bevano Rivers. The simulated elevation of the river water is also negative at all the time in Savio and Marecchia (Figure 10). The rest of the rivers such as the Rubicone, Uso and Marano Rivers, show positive elevation of river water. The Bevano River often reaches 0.04 m (Table 1), close to 0 m, and must be considered as the zone to allow seawater intrusion. By comparing the slope of the river bed, ranging from 0.001 to 0.0009, the

Marecchia River is considered less vulnerable than the other two rivers, i.e., the Savio and Bevano Rivers.

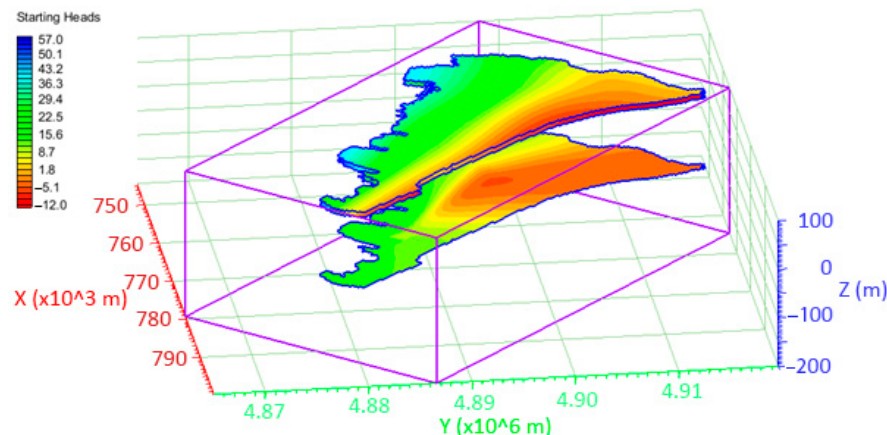

**Figure 9.** Three-dimensional distribution of groundwater table and piezometric head in A0, A1 and A2 aquifers.

**Figure 10.** Simulated elevation of river water, flow volume and thickness of water column at the cross-section next to the coastal boundary.

## 5. Summary and Conclusions

In the southern Emilia Romagna region, lowland phreatic aquifers are characterized by high vulnerability, being directly related with the surface water bodies and surface channels throughout the plain, as well as with the sea in the coastal area.

The present study regards the conceptualization and steady-state calibration of the multilayer aquifer system of the southern coast of Emilia Romagna.

The acceptable values of statistical parameters indicate that the calibration of the present multilayer model is reliable with a good match between observed and simulated groundwater table and piezometric heads. In comparison, the match is better in the shallow unconfined A0 aquifer than in the deep-seated A2 aquifer. The collection of additional piezometric data from the A2 aquifer may help to improve accuracy.

The calibrated model in the present study is preliminary and is subjected to many assumptions regarding the investigation of the field condition, as suggested by The Water Framework Directive (2000/60/EC).

The developed model is a preparatory step for the hydrogeological modeling of the salinization of the aquifers due to the seawater intrusion phenomenon. Therefore, future developments will concern the implementation of a density-dependent groundwater flow model through the SEAWAT code.

Naturally, a detailed description of the full density field in the area under investigation is costly and difficult to obtain. At minimum though, all head measurements should be carried out in conjunction with measurements of electrical conductivity. The density can then be estimated from simple relationships between density and salinity available in the literature [57].

The groundwater is exploited from the shallow aquifer and the deep-seated aquifers.

Since the groundwater gradient is flat at the majority of the inland area and grading downward from the sea to inland at the coastal area, the developed model can be used to simulate the impacts of overexploitation of the groundwater system. Due to the high vulnerability of all the aquifers to salinization, the developed model calibration can be refined using a three-dimensional distribution of salinity on the basis of density contrast. It is also reported that the aquifers show connectivity between each other along the alluvial fan deposits at the foot hills, at the NE part of the study area and at selected spots at the inland. The connectivity between the aquifers could be further investigated in relation to seawater intrusion problems. In addition, the developed model can also be used to quantify the role of surface water features and artificial drains in the shallow unconfined aquifer.

The present steady-state flow model can, therefore, be used as a tool to explore the vulnerability to seawater intrusion under different pumping scenarios, taking sea level rise (SLR) into account, one of the major climate-change-induced risks. Communicating the results to end users, authorities and policymakers will be helpful in formulating sustainable water management in the given sensitive and more dynamic complex aquifer system.

**Author Contributions:** S.S. was involved in the data analysis, model development and prepared the draft copy of the manuscript. C.C. was involved in the model conceptualization, technical discussion, review and supervision of the presented work. N.P. was involved in the model conceptualization, technical discussion, review and finalization of the manuscript. All authors have read and agreed to the published version of the manuscript.

**Funding:** This research received no external funding.

**Data Availability Statement:** Not applicable.

**Acknowledgments:** The authors wish to acknowledge MIUR Funds, Department of Excellence LP4-CUP F71G18000210001 for the research grant.

**Conflicts of Interest:** The authors declare no conflict of interest.

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
