# Peer review of "Dynamics of Coastal Aquifers: Conceptualization and Steady-State Calibration of Multilayer Aquifer System—Southern Coast of Emilia Romagna"

_water, doi:10.3390/w15132384_

Round 1

Reviewer 1 Report

Lines 314-315 pertaining to eq. (1), versus model construction in figure 6:
how can we know that the permeability tensor’s principal axes stay the same over the entire model domain, since the overall tectonic setting and surface hydrology phenomena don’t seem to globally favor the unique coordinate system (x, y, z) shown in figure 9, but rather suggest (figure 2) that the principal axes might rotate, at aquifer scale, both horizontally and vertically?
 – Well, I agree that such questions cannot be answered in practice, thus some assumptions and simplifications must be accepted.

Please check axes’ label and tick mark sizes in the r.-h.s. column of figure 10 for legibility in printed-paper format, and in figure 9 for the Z-axis (just for aesthetics’ sake).

In figure 10, I guess
“Water stage” stands for “Water level”
and
“Flow volume” stands for “Flow rate”.

The manuscript is well-written; however it would benefit from one final proofreading (by the authors themselves), to fix minor inadvertences in punctuation and grammar: missing commas, misplaced commas, missing hyphens, subject/verb agreement issues, like
– “no flow boundary” → “no-flow boundary”
– “steady state flow model” → “steady-state flow model”
– line 510: “relationship that exist”
– line 645 “represents”
    (you mean something like
     “accounts for the fact”,
       or rather “indicates”, “reveals”?)
– line 743: “model can be calibrated more”
     (suggest: “model calibration can be refined”)
– line 747: “as far as … ” (elliptic sentence)
– line 751-752: missing period at end of sentence
                         and/or elliptic sentence?

Author Response

The authors thank reviewer 1 for the valuable comments and review. The manuscript has been carefully and accurately revised. 

Reviewer 2 Report

This manuscript represents a case study of groundwater modeling in multi-layer aquifer. The major comments are as follows,

(1) As the study site is sea-side, it is not reasonable to choose MODFLOW as the model, since it cannot be used for variable porosity fluid. Why did you choose MODFLOW while not SEAWAT?

(2) It is not relevant to mention so much about the impact of sea level rise, climate change in Introduction part, since you have not investigate the topics. It is too long and make no sense in Introduction part. Authors should focus on their own work.

(3) I did not found the validation results, which I would think is necessary. Why could you not split the dataset as two, one for calibration and another for validation?

(4)Why did you set the flow as steady state flow? Even the results in Figure 4 shows the annual value change in different years.

(5)Are the result shown in Table 3 mathched with those in Figure 7? Why did you split them in two different parts?

(6) I would suggest to re-organize the paper to include the different senarios simulation to expand the contents of the paper. Current form did not include enought contents, since only the model calibration result was shown.

(7) Figures should be re-drawn. Units should be added in tables and figures.

(8) I do not agree with that this model can be used for sea water intrusion simulation shown in conclusion part as it cannot handle with sea water flow. 

(9) The innovative part of this manuscript is lack.

Author Response

The authors thank reviewer 2 for the valuable comments and review. The manuscript has been carefully and accurately revised. 
